# Peroxymonosulfate Activation by Facile Fabrication of α-MnO_2_ for Rhodamine B Degradation: Reaction Kinetics and Mechanism

**DOI:** 10.3390/molecules28114388

**Published:** 2023-05-27

**Authors:** Juexiu Li, Qixu Shi, Maiqi Sun, Jinming Liu, Rui Zhao, Jianjing Chen, Xiangfei Wang, Yue Liu, Weijin Gong, Panpan Liu, Kongyao Chen

**Affiliations:** 1School of Energy & Environment, Zhongyuan University of Technology, Zhengzhou 450007, China; 2College of Chemistry, Zhengzhou University, Zhengzhou 450001, China; 3International Education College, Henan Agricultural University, Zhengzhou 450002, China; 4School of Ecology & Environment, Zhengzhou University, Zhengzhou 450001, China; 5Henan Key Laboratory of Functional Salt Materials, Zhongyuan University of Technology, Zhengzhou 450007, China

**Keywords:** peroxymonosulfate, advanced oxidation process, manganese dioxide, degradation, Rhodamine B

## Abstract

The persulfate-based advanced oxidation process has been an effective method for refractory organic pollutants’ degradation in aqueous phase. Herein, α-MnO_2_ with nanowire morphology was facially fabricated via a one-step hydrothermal method and successfully activated peroxymonosulfate (PMS) for Rhodamine B (RhB) degradation. Influencing factors, including the hydrothermal parameter, PMS concentration, α-MnO_2_ dosage, RhB concentration, initial pH, and anions, were systematically investigated. The corresponding reaction kinetics were further fitted by the pseudo-first-order kinetic. The RhB degradation mechanism via α-MnO_2_ activating PMS was proposed according to a series of quenching experiments and the UV-vis scanning spectrum. Results showed that α-MnO_2_ could effectively activate PMS to degrade RhB and has good repeatability. The catalytic RhB degradation reaction was accelerated by increasing the catalyst dosage and the PMS concentration. The effective RhB degradation performance can be attributed to the high content of surface hydroxyl groups and the greater reducibility of α-MnO_2_, and the contribution of different ROS (reactive oxygen species) was ^1^O_2_ > $O_{2}^{\cdot -}$ > ${\mathrm{SO}}_{4}^{\cdot -}$ > ·OH.

## 1. Introduction

Dye wastewater, as a typical refractory organic wastewater, mainly emitted from the printing and textile industry, is characterized by toxicity, high chroma, and chemical oxygen demand (COD) [1]. The complex organic dyes with an aromatic structure may pose a serious threat to the water ecosystem and human health, with negative effects including aesthetic quality, photosynthetic activity, light penetration, and gas solubility [2,3]. Therefore, advancing dye wastewater treatment has drawn much attention. Conventional treatment technology for the elimination of organic pollutants can be achieved by filtration [4], adsorption [5], precipitation [6], and biological treatment [7]. However, these methods usually have the disadvantages of low efficiency and throughput requirements, which may also require further treatment of sorbent waste and membrane concentrate streams, and were inferior to meet deep organic components’ degradation and mineralization demand [8,9,10].

Advanced oxidation processes (AOPs) have been continuously developed since they were initially proposed by Hoigné and Bader (1976) [11] and Glaze (1987) [12]. A variety of reactive oxygen species (ROS) can be generated by oxidants under light irradiation [13], electricity [14], microwave [15,16], ultrasound [17], high temperatures [18], high pressure [19], etc. The targeted organic molecules can be effectively degraded into non-toxic inorganic salts, CO_2_ and H_2_O [14,20,21,22]. Among the emerging AOPs, persulfate-based oxidation processes utilize peroxydisulfate (PDS) and peroxymonosulfate (PMS) as major oxidants to induce ROS, featuring a higher oxidation potential (2.5–3.1 V), longer half-lifespan (3–4 × 10^−5^ s), wider pH operation range (2–8), cheap cost, and easy storage [23,24,25,26]. PDS and PMS exhibit differences in the activation pathway and nonradical reactions due to their structural differences. In comparison with PDS (^−^O_3_SO-OSO_3_^−^), the PMS molecule is asymmetrical (^−^O_3_SO-OH), with the attached sulfite (SO_3_) on one side and hydrogen on the other side. Therefore, PMS is more active than PDS owing to the nucleophilic attacks by organic pollutants with electron-rich moieties [27]. The ROS produced by activating persulfate mainly include powerful oxidants such as sulfate radical (${\mathrm{SO}}_{4}^{\cdot -}$), hydroxyl radical (OH), superoxide radical ($O_{2}^{\cdot -}$), and singlet oxygen (^1^O_2_). Among them, non-selective ·OH can effectively degrade and mineralize various organic pollutants in a wide range of pH values [28]. Additionally, various persulfate activation methods are developing, such as physical (ultraviolet, ultrasound, heating), chemical (electrical, heterogenous catalysis), and combinations of multiple methods [29,30]. Of these, persulfate activation by introducing homogeneous and heterogeneous catalysts are more promising due to their high activation efficiency, low operation costs, and mild operating conditions, without external energy [31].

Numerous studies have been carried on different persulfate activation materials, including transitional metal ions (Mn^2+^, Co^2+^, Cu^2+^, Fe^2+^) [32,33], transitional metal oxides [34], perovskite [35], and spinel catalysts [36]. Among them, transition metal oxides are excellent candidates in environmental applications in catalytic ozonation, photocatalysis, and VOCs’ oxidation [37,38], due to their wide sources and variable valence states. Manganese oxides (MnO, MnO_2_, Mn_2_O_3_, Mn_3_O_4_) are non-precious and have a high natural abundance, and they have been selected as promising candidates for the catalytic persulfate activation [39]. Especially, MnO_2_ has various crystal phases (α, β, γ, and δ), versatile morphologies (nanosheets, nanorods, nanotubes, and nanowires), different structures (tetragonal, hexagonal pore channel, and symmetrical cubic structures), and diverse exposed crystal planes, which can influence the physicochemical properties and further affect the catalytic performance of PS activation [21]. For example, Yang et al. [40] fabricated α-MnO_2_ with {110} and {100} as exposed facets, and successfully activated PMS for the degradation of Orange I (OI). Wang et al. [41] investigated different crystal forms of MnO_2_ (α-MnO_2_, β-MnO_2_, γ-MnO_2_, and δ-MnO_2_) in activating PMS for the removal of Reactive Yellow X-RG.

In this study, α-MnO_2_ with nanowire morphology were prepared using the facile hydrothermal method with different manganese precursor ratios and heating times. The as-prepared α-MnO_2_ were further characterized by X-ray diffraction (XRD), Brunauer–Emmett–Teller (BET), Fourier transform infrared (FTIR), field-emission scanning electron microscopy (SEM), and electrochemical analysis methods. Rhodamine B (RhB), a typical azo dye, was taking as target pollutant and thus simulating the refractory oraganic wastewater. The catalytic PMS activation performance was investigated under ambient temperature at 25 ± 2 °C. Influencing factors, including the manganese precursors ratio, hydrothermal time, PMS and α-MnO_2_ dosage, RhB initial concentration, pH value, and anions, were systematically investigated and discussed. The pseudo-first-order kinetic was further fitted to evaluate the corresponding reaction kinetics. The RhB degradation mechanism was also discussed based on scavenger tests and UV-vis scanning results.

## 2. Results and Discussion

### 2.1. Catalysts’ Characterization Result

The crystal phase and crystallinity of the as-prepared α-MnO_2_ were confirmed by XRD analysis, as shown in Figure 1a. It can be clearly seen that all the α-MnO_2_ by different hydrothermal times and precursor ratios exhibited a high diffraction peak intensity, indicating the good crystallization of α-MnO_2_ by using hydrothermal synthesis. Compared with the standard PDF, the diffraction peaks on the XRD pattern of all samples were well-matched to α-MnO_2_ (JCPDS: 44-0141). Specifically, the diffraction peaks at 2θ = 12.67°, 17.94°, 28.66°, 37.50°, 41.94°, and 49.79° were observed in the curves for all the catalysts, and these peaks were corresponding to the (110), (200), (310), (211), (301), and (411) crystal faces of α-MnO_2_. The results suggest that the crystal structure of α-MnO_2_ will not be affected by either the Mn^7+^/Mn^2+^ ratios in the precursor or the hydrothermal time.

To evaluate the surface area and the pore size distribution, α-MnO_2_ was examined by N_2_ adsorption/desorption measurements, as shown in Figure 1b. The longer hydrothermal reaction led to a decreased trend of the surface area, with α-MnO_2_-6 (1:1) of 50.6 m^2^/g, α-MnO_2_-6 (2:3) of 44.8 m^2^/g, α-MnO_2_-12 (1:1) of 35.4 m^2^/g, and α-MnO_2_-12 (2:3) of 43.1 m^2^/g. To obtain the surface functional group, α-MnO_2_ was analyzed by FTIR, as shown in Figure 1c. The main absorption peaks at 3440 and 1626 cm^−1^ were attributed to α-MnO_2_ surface hydroxyl groups’ stretching and bending vibrations. The peaks at approximately 1121, 716, and around 500 cm^−1^ were attributed to the stretching vibration of the Mn–O and Mn–O–Mn bonds. The redox performance of α-MnO_2_ was also preliminarily analyzed by electrochemical tests. Specifically, the powder α-MnO_2_ catalyst was inherent to the titanium plate and was used as a working electrode, and RuO_2_-IrO_2_/Ti was used as an auxiliary electrode. As shown in Figure 1d, a higher current density and a greater reductive ability were observed in α-MnO_2_-12 (2:3) and α-MnO_2_-12 (1:1), compared with that of α-MnO_2_-6.

The surface morphology of α-MnO_2_ at different hydrothermal times is shown in Figure 2. Apparently, α-MnO_2_ fabricated under 6 and 12 h hydrothermal reactions showed similar nanowires. The enhancement of the hydrothermal reaction time resulted in little influence on the α-MnO_2_ morphology. Moreover, the diameter of 6 h-1:1 was smaller than that of 12 h-1:1, indicating a larger surface area, corresponding to the BET results in Figure 1b. Specifically, the diameter of 6 h-1:1 α-MnO_2_ was 30~50 nm, with an average length of 1.5~2.0 μm. After the 12 h hydrothermal reaction, the α-MnO_2_ average diameter and length increased to 40~80 nm and 2.0~4.0 μm, respectively. Through the element mapping analysis, it was observed that oxygen, potassium, and manganese were uniformly dispersed in α-MnO_2_ particles, thus providing abundant active sites for PMS activation.

### 2.2. α-MnO_2_ Catalytic PMS Activation Performance

#### 2.2.1. Effect of Hydrothermal Preparation Parameters

The influence of different catalyst preparation parameters was studied and depicted in Figure 3a. Under the PMS alone condition, the RhB degradation process was not obvious within 30 min, which may be attributed to the weak PMS oxidation capacity towards RhB without further activation. When only α-MnO_2_ was added in RhB solution, the RhB degradation efficiency varied from 2.7% to 40.6% within 30 min for different α-MnO_2_. It can be inferred that the as-prepared nanowire α-MnO_2_ with a surface area of 35.4~50.6 m^2^/g exhibited adsorption and a slight catalytic degradation ability of RhB [42]. For α-MnO_2_-6 (1:1) and α-MnO_2_-6 (2:3), the RhB degradation efficiency reached 95.92% and 91.12% after 30 min. When the preparation time of the catalyst increased to 12 h, the RhB degradation efficiency was 88.87% for α-MnO_2_-12 (1:1) and 90.43% for α-MnO_2_-12 (2:3). The degradation kinetics was fitted based on the preliminary 20 min data of the degradation curve, and the results are shown in Figure 3b, which were coherent with the pseudo-first-order reaction. When the molar ratio of KMnO_4_ and MnSO_4_·H_2_O was 1:1 and 2:3 for the 12 h hydrothermal reaction, the first-order rate constant (K_obs_) of the RhB removal rate was 0.0718 and 0.0779. For the 6 h hydrothermal reaction, when the molar ratio of KMnO_4_ and MnSO_4_·H_2_O was 1:1 and 2:3, the reaction rate constant values of the RhB removal rate were 0.0918 and 0.0808, respectively. The above experiments showed that both the hydrothermal time and the precursor ratio had a certain effect on the degradation rate of MnO_2_-activated PMS. We selected α-MnO_2_-12 (1:1) for the following catalytic PMS activation experiment.

#### 2.2.2. Effect of PMS Dosage

The oxidant concentration determines the free radical quantity in the reaction system. The effect of different PMS amounts on catalytic RhB degradation performance was investigated. The PMS dosage was set at 0.01, 0.02, 0.05, 0.1, 0.2, and 0.5 g/L. As can be seen from Figure 4a, with the increase of the PMS concentration, the degradation efficiency of RhB significantly increased from 62.2% at 0.01 g/L to 94.4% at 0.5 g/L. As can be seen from Figure 4b, the removal rate of RhB fit well with the pseudo-first-order kinetic, with R^2^ values greater than 0.95. With the increase of the PMS concentration from 0.01 g/L to 0.5 g/L, the reaction rate constant value drastically increased from 0.037 min^−1^ to 0.123 min^−1^, indicating that the RhB degradation was strongly dependent on the PMS concentration. More reactive oxygen species was produced via the direct increase of the PMS concentration [43].

#### 2.2.3. Effect of α-MnO_2_ Dosage

The α-MnO_2_ catalyst dosage was also investigated, as shown in Figure 5. The number of reaction sites was determined by the catalyst concentration in the heterogeneous catalytic reaction [44]. As can be seen from Figure 5a, the degradation rate of RhB increased from 60.15% to 94.97% after 30 min, as the dosage of α-MnO_2_ increased from 0.05 g/L to 1 g/L. As shown in Figure 5b, the α-MnO_2_ dosage increased from 0.05 g/L to 0.1 g/L, 0.2 g/L, 0.5 g/L, and 1 g/L. The reaction rate constant value increased from 0.02 min^−1^ to 0.03 min^−1^, 0.05 min^−1^, 0.13 min^−1^, and 0.15 min^−1^, respectively. The improvement of the degradation rate can be attributed to the increasing number of PMS-activating sites provided by improving the α-MnO_2_ dosage, thus accelerating the production of reactive oxygen species.

#### 2.2.4. Effect of RhB Initial Concentration

The organic pollutants’ concentration is often variable in the effluent; therefore, the influence of the initial concentration of RhB should be considered. As shown in Figure 6a, the degradation effect of RhB decreased with the increase of the RhB concentration. When the initial concentration was 20 mg/L, 50 mg/L, and 80 mg/L, the degradation rates of α-MnO_2_ to RhB were 94.97%, 84.67%, and 78.31% after the 30 min reaction. The corresponding reaction rate constant values were 0.0779 min^−1^, 0.0643 min^−1^, and 0.0538 min^−1^, respectively (Figure 6b). The higher RhB initial concentration led to an increase in the RhB absolute amount in the solution, which requires more reactive oxygen species to react with the target pollutants and degradation intermediates. Under the same α-MnO_2_ and PMS dosages, the number of free radicals generated by catalytic activation was constant. Therefore, the increase of the RhB concentration led to a decrease of the degradation efficiency. However, the decolorization rate could remain at 78.31% with a corresponding RhB decolorization amount of 6.26 mg in 30 min, which exhibited a fascinating degradation performance compared to that of photocatalytic and biological methods when targeting high-chroma dye wastewater.

#### 2.2.5. Effect of pH Value and Cations

The pH value of a solution can change the surface charge distribution of Mn-based materials and affect the interaction and reactivity between the catalyst and PMS [45]. The effect of pH on the degradation of RhB in the α-MnO_2_ activating PMS system was also investigated. As can be seen from Figure 7a, when the solution pH varied from 3, 5, 7, and 9, the degradation rates of RhB after 30 min were 88.54%, 87%, 90.43%, and 84.2%, respectively. When the pH was 9, the RhB degradation efficiency was slightly inhibited. One reason may be that PMS was unstable under alkaline conditions, and the spontaneous decomposition of PMS might occur when high concentrations of hydroxide ions are present. In addition, the recombination of ${\mathrm{SO}}_{4}^{\cdot -}$ and ·OH to form ${\mathrm{HSO}}_{4}^{-}$ and oxygen may have occurred at high pH values [42,46]. Adjusting the pH value from 3 to 9 resulted in an overall slight influence on the RhB degradation efficiency, indicating that the α-MnO_2_ activating PMS system was suitable under a wide range of pH conditions. It should be noted that the pH value slightly decreased when PMS was added, and at the end of the reaction, the pH value remained at 2.91 to 3.86 when the initial pH was from 3 to 9.

Inorganic anions and natural organic matter (NOM) coexist in natural water and can influence the organic pollutants’ oxidation process depending on the competitive reaction of anions with different ROS [47]. Therefore, it is also necessary to investigate the effect of anions and NOM on RhB degradation in the α-MnO_2_ activating PMS system. Four common anions (${HCO}_{3}$^−^, Cl^−^, NO_3_^−^, and H_2_${PO}_{4}$^−^) and humic acid (HA) were added into the RhB solution before each test. As shown in Figure 7b, HA, NO_3_^−^, H_2_${PO}_{4}$^−^, and HCO_3_^−^ inhibited the RhB degradation efficiency to different extents, while adding 2 mM of Cl^−^ had little effect on RhB degradation: the degradation rate of RhB after the 30 min reaction reached 91.75%. The RhB degradation efficiency decreased to 47.35% at 30 min with 10 mg/L of HA. This can be explained by the fact that HA with abundant electron sites competed with RhB molecules for the electron-loving groups (${\mathrm{SO}}_{4}^{\cdot -}$ and ·OH) and the active sites (−OH) on the surface of MnO_2_ [48]. ${HCO}_{3}$^−^ and H_2_PO_4_^−^ greatly exhibited RhB degradation. When 2 mM of ${HCO}_{3}$^−^ was added, the degradation rate of RhB decreased from 90.43% to 16.5%, and owing to this, HCO_3_^−^ could react with ${\mathrm{SO}}_{4}^{\cdot -}$ and ·OH, followed by generation of carbonic acid with a weak oxidation ability, as shown in Equations (1)–(4) [48]. When 2 mM of H_2_${PO}_{4}$^−^ was added, the degradation of RhB was significantly inhibited, and the removal rate of RhB decreased from 90.43% to 31.97% within 30 min. This is because large amounts of ${\mathrm{SO}}_{4}^{\cdot -}$ and ·OH react with H_2_PO_4_^−^. ${NO}_{3}$^−^ could interact with the free electrons when PMS was activated and convert to NO_2_^−^, which could scavenge part of the ROS [49]. The RhB degradation rate slightly declined to 85.1% after inducement of 2 mM of NO_2_^−^.
(1)$${\mathrm{HCO}}_{3}^{-}+{\mathrm{SO}}_{4}^{\cdot -}\to {\mathrm{SO}}_{4}^{2-}+{\mathrm{HCO}}_{3}^{\cdot }$$
(2)$${\mathrm{HCO}}_{3}^{-}+\cdot \mathrm{OH}\to {\mathrm{OH}}^{-}+{\mathrm{HCO}}_{3}^{\cdot }$$
(3)$${\mathrm{CO}}_{3}^{2-}+{\mathrm{SO}}_{4}^{\cdot -}\to {\mathrm{SO}}_{4}^{2-}+{\mathrm{CO}}_{3}^{\cdot }$$
(4)$${\mathrm{CO}}_{3}^{2-}+\cdot \mathrm{OH}\to {\mathrm{OH}}^{-}+{\mathrm{CO}}_{3}^{\cdot }$$

#### 2.2.6. Radical Scavenger Tests

In order to unveil the degradation mechanism of RhB in the α-MnO_2_ activating PMS system, quenching experiments were conducted to identify the potential ROS contributions. Note that MeOH and TBA were employed as ${\mathrm{SO}}_{4}^{\cdot -}$ and ·OH scavengers, and p-BQ and FFA were chosen to quench $O_{2}^{\cdot -}$ and ^1^O_2_, respectively. As shown in Figure 8, after the 30 min reaction, the RhB degradation efficiency was inhibited by 2.62% and 7.59% when adding 0.1 M of TBA and MeOH, respectively. While the RhB degradation efficiency further decreased to 72.80% and 65.06%, the RhB degradation efficiency was inhibited by 17.64% and 25.34% by introducing p-BQ and FFA as quenchers, respectively. Based on the radical results, we can infer that all the above free radicals are involved in the degradation reaction, and the contributions to RhB degradation followed the order of: ^1^O_2_ > $O_{2}^{\cdot -}$ > ${\mathrm{SO}}_{4}^{\cdot -}$ > ·OH. $O_{2}^{\cdot -}$ and ^1^O_2_ were the main ROS for RhB degradation [40], and ^1^O_2_ may have also partly originated from PMS self-decomposition [50].

#### 2.2.7. Repeatability

The powder catalyst has certain limitations due to the separation and collection handling in industrial applications. Herein, we tested the repeatability of α-MnO_2_ for four cycling catalytic reactions. After each use, the catalyst was retained for direct next cycle reaction without further treatment. As can be seen from Figure 9, the degradation efficiency showed a declining trend with the increase of the cycle times overall, and the degradation efficiency of RhB in water did not significantly decrease after the initial two cycles. After three cycles, the RhB degradation efficiency decreased from 90.43% to 76.83%, which may be due to the loss of the catalyst in the cycling tests, the adsorption of RhB degradation intermediates, and the leaching of manganese ions in activating the PMS process. The cycling tests indicated that the as-prepared α-MnO_2_ has good reusability.

### 2.3. RhB Degradation Mechnism

The UV-visible spectra of the reactive solution in α-MnO_2_ activating PMS degradation of RhB at different reaction times is shown in Figure 10. It was apparent that all major absorbance peak intensities rapidly decreased with the increasing reaction time, indicating that the RhB molecule gradually degraded. The main absorption peak of the solution at 553 nm sharply weakened, and finally, there was no absorption peak intensity. When the catalytic reaction time was 25 min, the chroma of the RhB solution became colorless, indicating that the chromophoric group had been destroyed. The reduction of the peaks at 553 nm mainly resulted from the breakage of the conjugated structure of C=N and C=O groups, which was responsible for the fading color of RhB. Based on the TOC measurement, the TOC value decreased from 3.18 mg/L to 0.304 mg/L after the 40 min reaction. Then, 90% of TOC was removed during RhB degradation, indicating that the intermediate degradation products were mineralized.

### 2.4. Possible Activation Mechanism

Based on the results, α-MnO_2_ activating PMS of the RhB degradation mechanism was proposed and depicted in Figure 11. Firstly, HSO^−5^ in solution was attached on the surface of α-MnO_2_. Different valences (Mn^2+^, Mn^3+^, and Mn^4+^) and surface −OH on the α-MnO_2_ surface acted as active sites to react with ${\;\mathrm{HSO}}_{5}^{-}$, thus generating sulfur-based radicals (${\mathrm{SO}}_{5}^{\cdot -}$ and ${\mathrm{SO}}_{4}^{\cdot -}$) (Equations (5)–(8)) [25], of which, a portion of ${\mathrm{SO}}_{4}^{\cdot -}$ would convert into ·OH through Equations (9) and (10) [21]. Secondary radicals, $O_{2}^{\cdot -}$, were generated via Equations (11)–(13), and the non-radical ^1^O_2_ was produced via Equations (14) and (15) [51]. All the above ROS contributed to the RhB degradation process (Equation (16)).
(5)$${\mathrm{Mn}}^{4+}+{\;\mathrm{HSO}}_{5}^{-}\to {\mathrm{Mn}}^{3+}+{\mathrm{SO}}_{5}^{\cdot -}+H^{+}$$
(6)$${\mathrm{Mn}}^{3+}+{\;\mathrm{HSO}}_{5\;}^{-}\to {\mathrm{Mn}}^{4+}+{\mathrm{SO}}_{4}^{\cdot -}+-\mathrm{OH}\;$$
(7)$${\mathrm{Mn}}^{2+}+{\mathrm{HSO}}_{5}^{-}\to {\;\mathrm{Mn}}^{3+}+{\mathrm{SO}}_{5}^{\cdot -}+-\mathrm{OH}$$
(8)$${\mathrm{Mn}}^{3+}+{\mathrm{HSO}}_{5}^{-}\to {\mathrm{Mn}}^{2+}+{\mathrm{SO}}_{5}^{\cdot -}+H^{+}$$
(9)$${\mathrm{SO}}_{4}^{\cdot -}+H_{2}O\;\to \cdot \mathrm{OH}\;+{\mathrm{SO}}_{5}^{2-}+H^{+}$$
(10)$${\mathrm{SO}}_{4}^{\cdot -}+-\mathrm{OH}\;\to \cdot \mathrm{OH}\;+{\;\mathrm{SO}}_{4}^{2-}$$
(11)$${\mathrm{HSO}}_{5}^{-}+H_{2}O\;\to H_{2}O_{2}+{\;\mathrm{HSO}}_{4}^{-}$$
(12)$$H_{2}O_{2}+\cdot \mathrm{OH}\;\to {\;\mathrm{HO}}_{2}^{\cdot }+{\;H}_{2}O\;$$
(13)$${\mathrm{HO}}_{2}^{\cdot }+H^{+}\to O_{2}^{\cdot -}$$
(14)2O2·-+ H+→O21+H2O2
(15)·OH →O21+-OH
(16)O21/ O2·-/ SO4·-/·OH+RhB → intermediates →CO2 +H2O

## 3. Materials and Methods

### 3.1. Chemicals

All chemicals were of analytical grade and used directly without further purification, unless otherwise specified. α-MnO_2_ were fabricated using KMnO_4_ and MnSO_4_·H_2_O as precursors. Peroxymonosulfate (2KHSO_5_·KHSO_4_·K_2_SO_4_) and Rhodamine B (RhB) were bought from Macklin Biochemical Technology and used as received. Potassium chloride (KCI), sodium bicarbonate (NaHCO_3_), and monopotassium phosphate (KH_2_PO_4_) were used to investigate the cation influence of the RhB degradation efficiency. The radical scavengers used in this study included tert-Butanol (TBA), p-Benzoquinone (p-BQ), methanol (MeOH), and furfuryl alcohol (FFA), respectively. The pH value was adjusted using sulfur acid solution and sodium hydroxide.

### 3.2. Catalysts’ Preparation

The α-MnO_2_ catalysts with nanowire morphology were synthesized via the facile hydrothermal method. Specifically, stoichiometric amounts of KMnO_4_ and MnSO_4_·H_2_O with Mn^7+^:Mn^2+^ ratios of 1:1 and 2:3 were dissolved in 80 mL of deionized water under vigorous stirring for 30 min. The obtained solid-solution mixture was further transferred to a 100 mL hydrothermal kettle made of Teflon liner and heated at 150 °C for 6 and 12 h. Ac-cording to the precursors’ different molar ratios of KMnO_4_ and MnSO_4_·H_2_O, the prepared α-MnO_2_ was denoted as α-MnO_2_-6 (1:1), α-MnO_2_-6 (2:3), α-MnO_2_-12 (1:1), and α-MnO_2_-12 (2:3), respectively. After cooling to room temperature, a solid material was obtained after filtration of the solid-solution mixture, washed, and dried at 80 °C overnight.

### 3.3. Analytical Instruments

The crystal phase of the prepared α-MnO_2_ crystal structure was determined by powder X-ray diffraction (XRD-6100, Shimadzu, Kyoto, Japan), using Cu Ka as a radiation source and a scanning range of 10–90° at a scanning speed of 5°·min^−1^. The N_2_ adsorption and desorption curve of α-MnO_2_ was evaluated over an Autosorb-iQ instrument (Quantachrome Corporation, Boynton Beach, FL, USA). Before testing, the sample was degassed at 300 °C for 6 h, and then the N_2_ adsorption/desorption test was conducted under liquid N_2_ at 196 °C. The specific surface area of the catalyst was determined using the multi-point Brunauer–Emmett–Teller (BET) method. The surface morphology of the sample was evaluated by a field-emission scanning electron microscope (FESEM: JEOL JSM-7800F, Tokyo, Japan) and the surface elements’ composition (Mn, K, and O) was surveyed via energy-dispersive spectroscopy (EDS). Fourier transform infrared (FTIR) spectra were recorded on a Nicolet 6700 spectrometer (Thermo Fisher, Waltham, MA, USA). The electrochemical measurements were carried out on a CHI660E electrochemical workstation.

### 3.4. Experimental Procedure

RhB degradation experiments by α-MnO_2_ activating PMS were conducted in a beaker containing 100 mL of RhB solution under continuous stirring at ambient temperature (25 ± 2 °C). Catalyst was initially added to RhB solution for 20 min to achieve saturated adsorption before the inducement of PMS. Samples of the RhB degradation solution were further withdrawn at 5 min intervals using a syringe and analyzed using the DR6000 UV-vis Spectrophotometer (Hach, Ames, IA, USA). The decolorization of RhB was investigated by measuring the maximum absorbance at a wavelength of 553 nm and by computing the concentration from the calibration curve. In order to avoid the adsorption influence of the dye color via membrane filtration, the RhB and degradation samples of 1 mL were withdrawn using a 5 mm optical cell and further analyzed without filtration after quenching with an excess amount of Na_2_S_2_O_8_. RhB absorption spectra were recorded at different intervals at a scanning wavelength range of 200–800 nm to analyze the degradation process. The degradation rate (*η*) of RhB and the pseudo-first-order reaction kinetics were calculated as follows:*η* = C/C_0_ × 100%(17)
ln(C/C_0_) = −kt(18)
where C and C_0_ are the RhB concentration at the sampling time and the initial concentration of RhB (mg/L), according to the absorbance at 553 nm, k is the pseudo-first-order reaction rate constant (min^−1^), and t is the reaction time (min).

In order to investigate the role of reactive oxygen species (ROS) in the catalytic system, methanol (MeOH), p-Benzoquinone (p-BQ), tert-butanol (TBA), and furfuryl alcohol (FFA) were used as ${\mathrm{SO}}_{4}^{\cdot -}$, $O_{2}^{\cdot -}$, ·OH, and O21 scavengers, respectively [8,40]. In the repeatable experiment, the catalyst remained for the direct next cycle reaction, without further treatment after each use.

## 4. Conclusions

In this study, α-MnO_2_ was successfully prepared via a hydrothermal method, which exhibited higher catalytic activity in PMS activation to degrade RhB. The influencing factors, including α-MnO_2_ preparation, PMS and α-MnO_2_ dosage, RhB initial concentration, pH value, and anions, were systematically investigated. The as-prepared α-MnO_2_ with nanowire morphology exhibited good crystallization, with little affection of the hydrothermal time and the precursor ratio. The effective RhB degradation performance can be attributed to the high content of surface hydroxyl groups and the greater reducibility of α-MnO_2_. More reactive oxygen species was produced via the direct increase of α-MnO_2_ and the PMS concentration. The α-MnO_2_ activating PMS process is also capable of high-chroma dye wastewater treatment and has a wide pH value range. After four cycling times, the RhB degradation retained good stability. The main ROS that participated in RhB degradation followed the contribution trend of:O21 > $O_{2}^{\cdot -}$ > ${\mathrm{SO}}_{4}^{\cdot -}$ > ·OH.

## Figures and Tables

**Figure 1 molecules-28-04388-f001:**
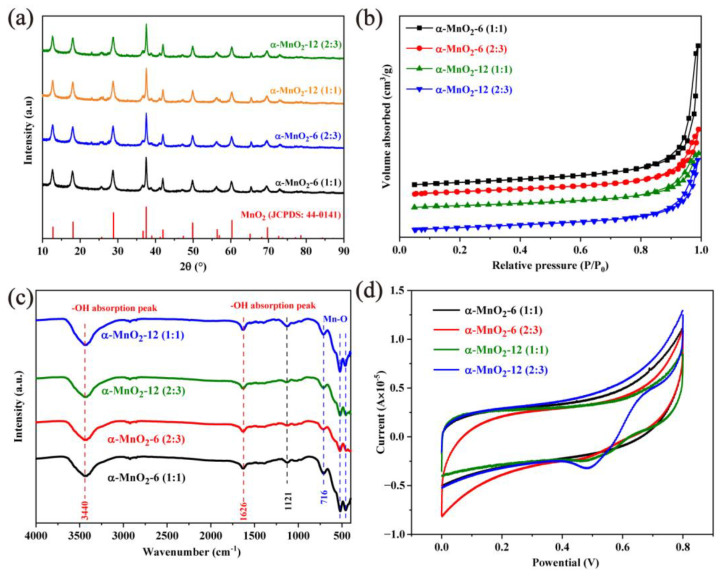
XRD patterns (**a**), N_2_ adsorption/desorption isotherm (**b**), FTIR spectra (**c**), and cyclic voltammetry curves, (**d**) of α-MnO_2_ under different preparation conditions.

**Figure 2 molecules-28-04388-f002:**
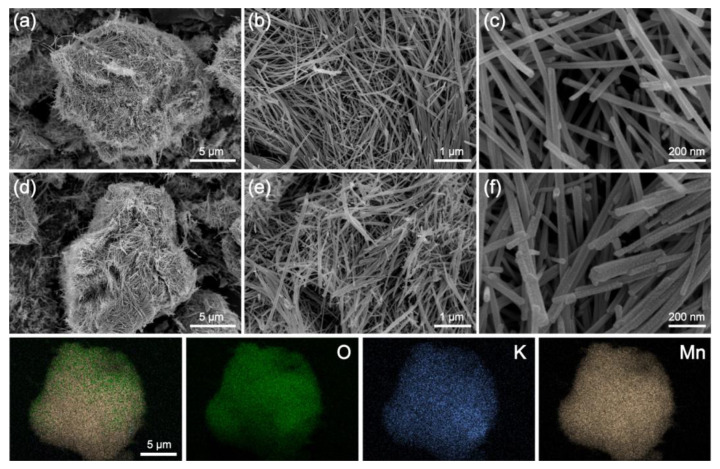
SEM images of α-MnO_2_-6 h 1:1 (**a**–**c**) and α-MnO_2_-12 h 1:1 (**d**–**f**), and elemental mapping results of α-MnO_2_-6 h 1:1.

**Figure 3 molecules-28-04388-f003:**
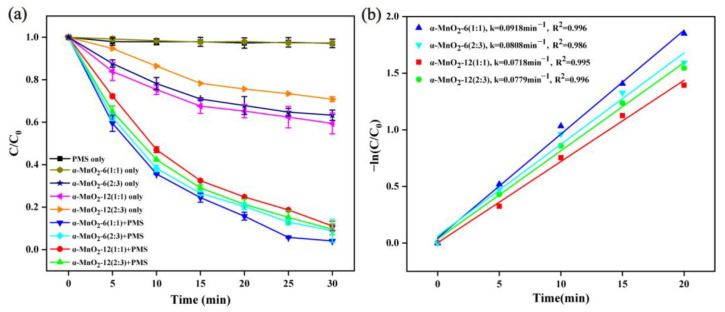
Effect of different α-MnO_2_ preparation parameters on RhB degradation (**a**) and corresponding pseudo-first-order kinetics (**b**) (reaction condition: α-MnO_2_ = 0.2 g/L, RhB = 20 mg/L, PMS = 0.05 g/L, initial pH = 7, temperature = 25 ± 2 °C).

**Figure 4 molecules-28-04388-f004:**
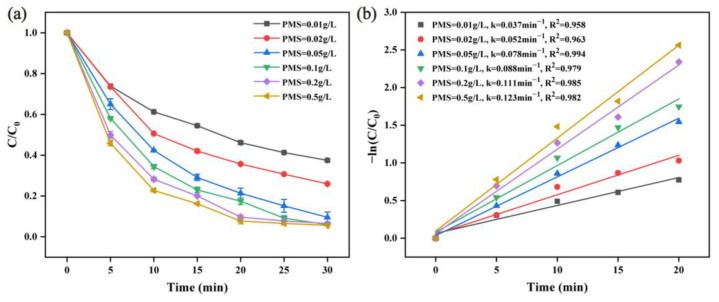
Effect of the PMS dosage on the RhB degradation efficiency (**a**) and the corresponding pseudo-first-order kinetic curves (**b**) (reaction condition: α-MnO_2_ = 0.2 g/L, RhB = 20 mg/L, initial pH = 7, temperature = 25 ± 2 °C).

**Figure 5 molecules-28-04388-f005:**
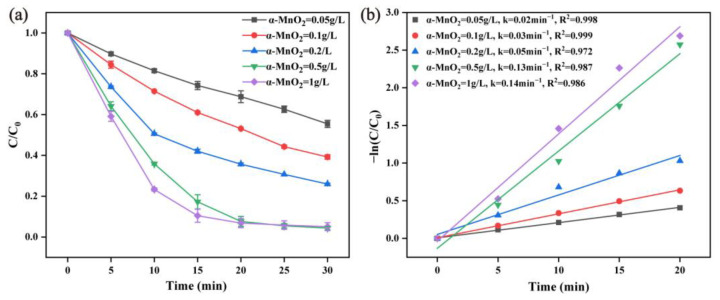
Effect of the α-MnO_2_ dosage on the RhB degradation efficiency (**a**) and the corresponding pseudo-first-order kinetic fitting curves (**b**) (reaction condition: PMS = 0.02 g/L, RhB = 20 mg/L, initial pH = 7, temperature = 25 ± 2 °C).

**Figure 6 molecules-28-04388-f006:**
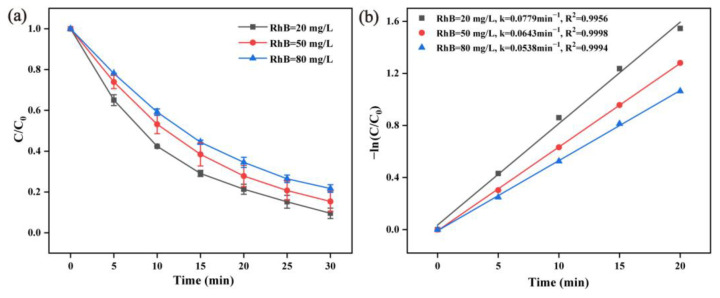
Influence of the RhB initial concentration (**a**) and the pseudo-first-order kinetic curves (**b**) (reaction condition: α-MnO_2_ = 0.2 g/L, RhB = 20 mg/L, PMS = 0.05 g/L, initial pH = 7, temperature = 25 ± 2 °C).

**Figure 7 molecules-28-04388-f007:**
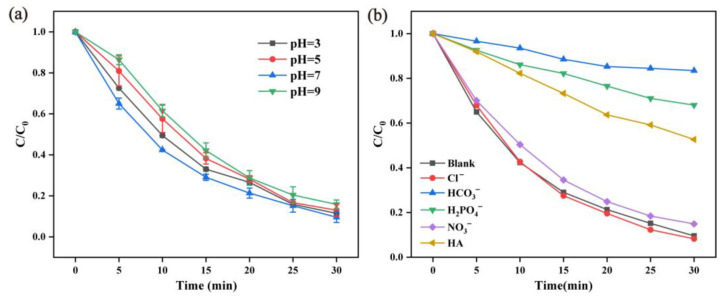
Influence of pH value (**a**) and cation (**b**) on the RhB degradation efficiency (reaction condition: α-MnO_2_ = 0.2 g/L, RhB = 20 mg/L, PMS = 0.05 g/L, temperature = 25 ± 2 °C).

**Figure 8 molecules-28-04388-f008:**
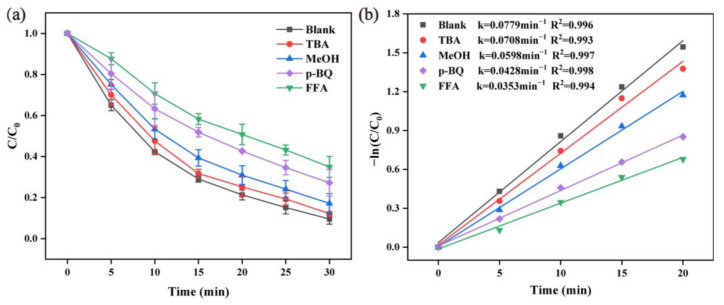
Influence of different radical scavengers (**a**) and the pseudo-first-order kinetic curves (**b**) (reaction condition: α-MnO_2_ = 0.2 g/L, PMS = 0.05 g/L, initial pH = 7, RhB = 20 mg/L, temperature = 25 ± 2 °C).

**Figure 9 molecules-28-04388-f009:**
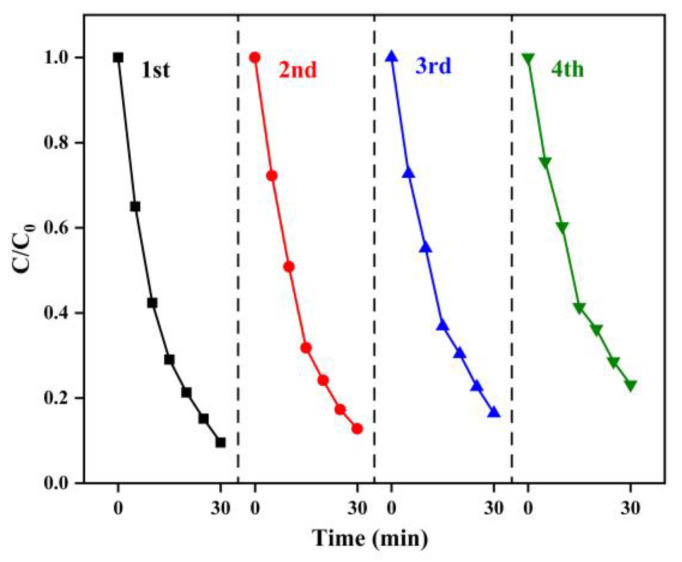
Repeatability of α-MnO_2_ for 4 cycling tests (reaction condition: α-MnO_2_ = 0.2 g/L, PMS = 0.05 g/L, initial pH = 7, RhB = 20 mg/L, temperature = 25 ± 2 °C).

**Figure 10 molecules-28-04388-f010:**
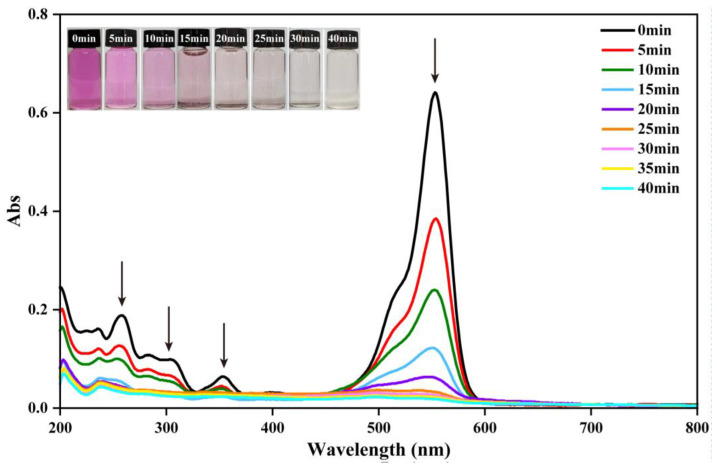
UV-vis scanning spectrum during the RhB degradation process (reaction condition: α-MnO_2_ = 0.2 g/L, PMS = 0.05 g/L, initial pH = 7, RhB = 20 mg/L, temperature = 25 ± 2 °C).

**Figure 11 molecules-28-04388-f011:**
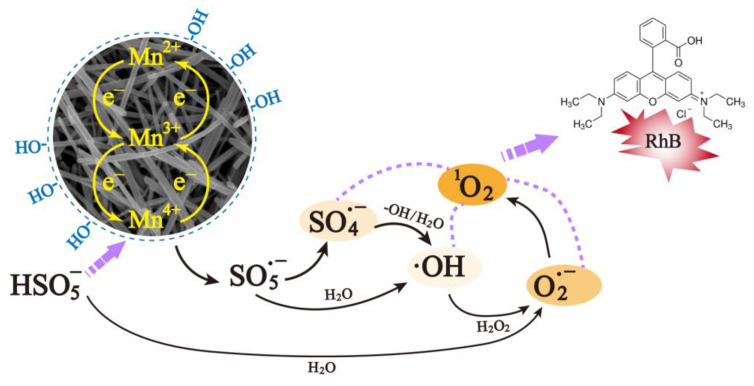
Proposed RhB degradation mechanism by α-MnO_2_ activating PMS.

## Data Availability

Data will be made available upon request from the corresponding author. The data are not publicly available due to privacy.

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
