# Peer review of "Peroxymonosulfate Activation by Facile Fabrication of α-MnO2 for Rhodamine B Degradation: Reaction Kinetics and Mechanism"

_molecules, 2023, doi:10.3390/molecules28114388_

Round 1
Reviewer 1 Report
(1) Line 18: Pls provide the full name of PMS.
(2) Line 21: “the” should be added in front of pseudo.
(3) Lines 40-42: pls add relevant references.
(4) Lines 48-49: Only electricity has cited, pls refer to 10.1016/j.cej.2014.03.086, 10.1016/j.cej.2017.01.012.
(5) The writing style of radicals should be corrected according to the following reference: “Koppenol, W. (2000). "Names for inorganic radicals (IUPAC Recommendations 2000)." Pure and Applied Chemistry 72(3): 437-446.”
(6) Lines 56-57: Only hydroxyl radical with nonselective can effectively degrade and mineralize various organic pollutants, pls revise.
(7) Some relevant references, such as 10.1016/j.cej.2020.128176 and 10.1016/j.cclet.2021.10.087 should be cited in Lines 63-67 in introduction.
(8) Line 68: Mn has high biotoxicity, maybe the expression of “environmental friendly” should be reconsidered.
(9) Line 95: KCl.
(10) Lines 128-130: More details, such as the quenchers or filters used, should be provided.
(11) Is it reliable to use spectrophotometer for RhB determination, especially in the presence of BQ, pls add some explanation?
(12) Figure 2b: Only N2 adsorption-desorption isotherm of two α-MnO2 is provided, pls revise.
(13) Figure 3a: the as-prepared nanowire α-MnO2 with different surface area should be tested for RhB absorption, pls revise.
(14) Lines 191-192: the not obvious degradation of RhB may be attributed to the weak oxidation ability of PMS toward RhB, pls revise.
(15) Section 3.2.5: Pls provide the pH values after the PMS addition and the end pH.
(16) What’s the influence of NOM?
(17) Figure 10: pls add labels for glass bottles.
(18) Lines 332-333: Pls add the TOC changes during the treatment.
(19) References style should be revised.
Author Response
Dear reviewer:
We are appreciated for the valuable comments and suggestions on our manuscript “Molecules-2360286” entitled “Peroxymonosulfate Activation by Facile Fabrication of α-MnO2 for Rhodamine B Degradation: Reaction Kinetics and Mechanism”. After carefully checking all the comments, we have done the revision according to the reviewers’ points step by step. Enclosed sentences with red mark are the point-to-point replies to the comments raised by the reviewer.

Reviewer 2 Report
The topic of the manuscript is well and fashionable - the combination of persulfate process with heterogeneous photocatalysis. However, the elaboration of the topic and the evaluation of the results contain many shortcomings, which is why I do not recommend the manuscript in its current form for publication.
1. The introduction needs to contain information and references about the possible reactive particles formed during PDS-heterogeneous photocatalysis combinations - this is an important, essential, and controversial area. But no information about this topic at all in the Introduction.
2. I consider it professionally wrong to build theory and conclusions on 2-3% efficiency differences (radical results).
3. Benzoquinone is not only an O2-radical scavenger in this case - the transformation of PDS/PMS can be catalyzed by semiquinone radicals.
4. A few sentences from the manuscript that, in my opinion, are unscientific, based on wrong information (or simply incomprehensible to me):
"The higher RhB initial concentration led to increasing RhB molecules in the solution,"
"As shown in Figure 7-b, NO-3, H2PO-4 and HCO-3 inhibited the RhB degradation efficiency to different extent, while Cl- promoted the degradation of RhB." - As I can see on Fig 7b NO-3 and Cl- has no effect..... (
What is the meaning of " mM/L" as unit?
. "When the pH was 9, RhB degradation efficiency was slightly inhibited because SO4·- was removed and the formation of ·OH was inhibited under alkaline conditions". Considering the first points of the kinetic curves, a systematic pH dependence can be observed - however, this is no longer observable after 20 minutes. What does "SO4·- was removed" mean? That is a well known fact, the hydrolysis of SO4·- produces ·OH above pH 9. Why authors think the formation of ·OH is hindered? What is the basis of this statement?
The application of radical scavengers, their expected effect, and the evaluation of the obtained results have not been evaluated correctly. (This part does not contain reaction rate constant values ​​at all, which is a serious shortcoming.) Concentration units are confusing (M/L, mM/L)
Overall, I think the topic of the manuscript is good, but its execution was rushed, the conclusions drawn from the results are not well-founded and it contains a lot of formal errors.
Author Response
Dear reviewer:
We are vary appreciated for the valuable comments and suggestions on our manuscript “Molecules-2360286” entitled “Peroxymonosulfate Activation by Facile Fabrication of α-MnO2 for Rhodamine B Degradation: Reaction Kinetics and Mechanism”. After carefully checking all the comments, we have done the revision according to the reviewers’ points step by step. Enclosed sentences with red mark are the point-to-point replies to the comments raised by the reviewer.

Round 2
Reviewer 1 Report
accept
Reviewer 2 Report
The authors have worked hard on the manuscript - the version uploaded in the first round really needed this. Although I do not agree with all the answers of the authors, the manuscript in its current form is suitable for publication.